# Deep Learning for Automated Ventricle and Periventricular Space Segmentation on CT and T1CE MRI in Neuro-Oncology Patients

**DOI:** 10.3390/cancers17101598

**Published:** 2025-05-08

**Authors:** Mart Wubbels, Marvin Ribeiro, Jelmer M. Wolterink, Wouter van Elmpt, Inge Compter, David Hofstede, Nikolina E. Birimac, Femke Vaassen, Kati Palmgren, Hendrik H. G. Hansen, Hiska L. van der Weide, Charlotte L. Brouwer, Miranda C. A. Kramer, Daniëlle B. P. Eekers, Catharina M. L. Zegers

**Affiliations:** 1Department of Radiation Oncology (Maastro), GROW Research Institute for Oncology and Reproduction, Maastricht University Medical Centre+, 6229 ET Maastricht, The Netherlands; marvin.ribeiro@maastro.nl (M.R.); wouter.vanelmpt@maastro.nl (W.v.E.); nikolina.birimac@maastro.nl (N.E.B.); femke.vaassen@maastro.nl (F.V.); rik.hansen@maastro.nl (H.H.G.H.); danielle.eekers@maastro.nl (D.B.P.E.); karen.zegers@maastro.nl (C.M.L.Z.); 2Department of Radiology and Nuclear Medicine, Mental Health and Neuroscience Research Institute (MHeNs), Faculty of Health Medicine and Life Sciences, Maastricht University, 6229 ER Maastricht, The Netherlands; 3Department of Applied Mathematics, Technical Medical Centre, University of Twente, 7522 NB Enschede, The Netherlands; j.m.wolterink@utwente.nl; 4Department of Radiation Oncology, University Medical Center Groningen, University of Groningen, 9713 AP Groningen, The Netherlands; h.l.van.der.weide@umcg.nl (H.L.v.d.W.); c.l.brouwer@umcg.nl (C.L.B.); m.c.a.kramer@umcg.nl (M.C.A.K.)

**Keywords:** deep learning, nnUNet, autocontouring, ventricles, periventricular space, neuro-oncology

## Abstract

In radiotherapy, it is important to minimize radiation to healthy tissue while delivering enough of the proper dose to the tumor. The region surrounding the brain ventricles, the periventricular space, seems particularly sensitive to radiation damage. This study aimed to develop and validate a deep learning model to automatically segment the ventricles and periventricular space on CT and MRI scans to improve treatment planning for patients receiving intracranial radiotherapy. The resulting model (nnU-Net) was tested alongside an existing model (SynthSeg) to see which performed better at segmenting the brain ventricles. The results showed that the new model, nnU-Net, performed more accurately and was preferred by radiotherapy technicians. These findings could improve the process of contouring organs at risk in brain cancer patients undergoing radiation therapy.

## 1. Introduction

Minimizing radiation to healthy brain tissue while delivering a sufficient radiation dose to tumors is an important balance to achieve during intracranial radiotherapy. Recent studies have shown that the 4 mm region surrounding the ventricles, the periventricular space (PVS), seems particularly sensitive to radiation damage, likely due to lower vascular supply and variability in biological damage when using particle therapy [1,2]. Due to its sensitivity to radiation, the PVS was officially classified as an organ at risk (OAR) in the 2021 EPTN Neuro-Oncology Atlas to limit radiation exposure to this region [3].

Because the PVS is included in the 2021 EPTN OAR list, its delineation is strongly recommended for all neuro-oncology patients treated with proton therapy in the Netherlands. However, manually contouring OARs is often time-consuming, particularly for larger, more complex structures such as the ventricles and by extension the PVS. Anatomical variations among patients and the narrow complex shape of the ventricles are often hard to detect in the available imaging, leading to interobserver variability [4]. Nevertheless, accurate and consistent PVS delineation is essential to assess its radiation sensitivity and to ensure dose sparing during treatment.

Given the increasing importance of accurate segmentation for treatment planning in neuro-oncology, there is a growing need for automated segmentation models that precisely segment the entire ventricular system. Deep learning (DL)-based segmentation methods offer a potential solution to the time-consuming and variable process of OAR delineation. Prior research has explored DL-based brain ventricle segmentation. However, these models are not optimized for neuro-oncology patients and fail to fully address the specific needs of radiotherapy. For instance, some models leave out essential structures of the ventricles by only include the lateral ventricles, leaving out the third and fourth ventricles [5,6]. Others are developed for different clinical contexts, such as hydrocephalus diagnosis and monitoring, where the emphasis is on ventricle parcellation rather than precise boundary delineation [7,8,9,10]. These models are typically trained on non-oncologic populations and optimized for different anatomical and diagnostic priorities. Furthermore, most prior studies have not incorporated the multimodal imaging pipelines (e.g., combining CT and T1 contrast-enhanced MRI) that are routinely used in radiotherapy workflows [11,12].

Creating a sufficiently large dataset suitable for training a DL model is another challenge due to the time-consuming task of data collection and manual delineation [13]. Pretrained off-the-shelf DL models may provide a quick solution for these problems. However, their accuracy and adherence to the EPTN guidelines may be insufficient for achieving clinically acceptable results within radiotherapy. These challenges highlight the need for automated solutions specifically tailored for radiotherapy applications.

This study had two main objectives. First, we aimed to develop a deep learning model for accurate automated segmentation of the ventricles and periventricular regions using CT and T1CE MRI in accordance with EPTN guidelines for radiotherapy planning. These guidelines allow the use of dose constraints on organs at risk (OARs), making precise segmentation clinically valuable. Second, we evaluated the accuracy and clinical applicability of this custom model compared to a publicly available pretrained model, using both internal and external test datasets. This study represents, to our knowledge, the first application of DL-based segmentation to support periventricular OAR delineation in neuro-oncology radiotherapy and provides insights into the capabilities and limitations of off-the-shelf DL solutions in this context.

## 2. Methods

### 2.1. Datasets

#### 2.1.1. Internal Training and Test Set

The dataset included 96 neuro-oncology patients treated with either photon (n = 24) or proton radiotherapy (n = 72) at the Department of Radiotherapy (Maastro) of Maastricht University Medical Center (MUMC) in Maastricht, the Netherlands, between 2019 and 2023. The dataset was collected retrospectively under Maastro institutional review board approval (project number P0632). Patient characteristics of this dataset are described in Table 1. For every patient, a CT scan (Siemens SOMATOM Drive and Confidence, Erlangen, Germany) with voxel dimensions of 0.68 × 0.68 × 1 mm and a high-resolution contrast-enhanced (gadolinium) T1-weighted 3T MRI (Philips Achieva, Best, The Netherlands) with voxel dimensions of 1 × 1 × 1 mm were available. The dataset encompassed a clinically complex population with a large variability in brain anatomy due to aging, neurodegeneration, hydrocephalus, resection cavities, and tumors. The dataset of 96 patients was randomly split into a training set of 78 patients and an internal test set of 18 patients. The selection of patients for the training and test sets was conducted manually. To minimize bias, the authors ensured that they were blind to any patient characteristics or imaging data during the selection process, thereby preventing any potential influence on the composition of the sets. Specifically, the dataset consisted of 96 patient folders, each containing a corresponding CT scan, MRI, and label set. Eighteen of these patient folders were manually and randomly selected to form the internal test set, with the remaining 78 used for training.

#### 2.1.2. External Test Set

An external test set was used to evaluate the performance of the model. The external test set included 18 neuro-oncology patients treated with proton therapy from the Department of Radiotherapy of the University Medical Center Groningen (UMCG). Patient characteristics of this dataset are described in Table 1. The external test set consisted of a CT scan (Siemens SOMATOM Definition AS, Erlangen, Germany) with voxel dimensions of 0.98 × 0.98 × 1 mm and a contrast-enhanced (gadolinium) T1-weighted MRI (Magnetrom Aera or Magnetrom AvantoFit Siemens, Erlangen, Germany), with voxel dimensions of 1 × 1 × 1 mm. The external test set consisted of a clinically complex population similar to the internal test set.

### 2.2. Ground Truth Delineations

Manual ventricle and PVS (defined as the 4 mm region surrounding the ventricles) delineations were performed on the CT scan using the registered T1-weighted MRI scan as an overlay. The EPTN 2021 delineation guidelines were used to contour both the ventricles and the PVS [3]. For the internal training and test set, delineations were performed by a radiotherapy technician and a researcher using Raystation 12A (RaySearch Laboratories AB) under the supervision of an experienced radiation oncologist. While not all segmentations were formally double-checked, difficult cases were reviewed in consensus with the supervising radiation oncologist to ensure delineation quality. Manual delineation of the ventricles took around 20 min per patient. All delineations in the external test set were performed by an experienced radiation oncologist using Raystation 11B.

### 2.3. Deep Learning Models

#### 2.3.1. nnU-Net for Deep Learning Framework

Convolutional neural networks (CNNs), particularly U-Net [14] and its automated version nnU-Net [15], have become highly effective in medical image segmentation, with nnU-Net offering an easy-to-use, state-of-the-art solution by automating key hyperparameter selection and preprocessing steps, maintaining high performance even in a recent study [16]. In this study, three different models were trained within the nnU-Net framework: (1) 2D U-Net; (2) 3D full-resolution U-Net; (3) 3D U-Net cascade, where the first 3D U-Net operated on low-resolution images, followed by a second, high-resolution 3D U-Net that refined the predictions of the first network.

Importantly, all preprocessing steps were fully handled by the nnU-Net pipeline without manual intervention. Default nnU-Net preprocessing steps included resampling to a standardized voxel spacing, intensity normalization, and spatial cropping. No additional manual preprocessing of the dataset was performed prior to training. The default nnU-Net architecture settings were used without any manual modifications to the number of layers, input patch size, or convolutional kernels. Data augmentation during training, including random rotations, scaling, elastic deformations, and intensity variations, was automatically applied through the nnU-Net’s default augmentation pipeline.

All three nnU-Net configurations were trained on the CT and T1CE MRI of the training set, with slight adjustments to the default nnU-Net settings. The T1CE MRI and CT of each case were used as input into a single trained model for each configuration. The standard hybrid loss function combining the DSC loss and Cross-Entropy loss was used. This combination leads to more robust segmentation performances by balancing their complementary strengths in handling class imbalance [17]. A stochastic gradient descent (SGD) optimizer with a learning rate of 0.01 and momentum of 0.99 was used. We reduced the number of training epochs from the default 1000 to 250, as preliminary experiments indicated that the model converged on our dataset within this reduced number of epochs. Given the limited size of the dataset, five-fold cross-validation was used during training for all three models to prevent overfitting and enhance the model’s robustness. The data splitting for the five-fold cross-validation was carried out using the nnU-Net framework. Specifically, a deterministic (seeded) randomization process was used to split the dataset into five folds, ensuring the splits were reproducible. After training, predictions were generated using an ensemble of the models from the five cross-validation folds. During inference, the nnU-Net utilized both the CT and T1CE MRI scans as input, reflecting its training process and the real-world clinical scenario where both modalities are available for radiotherapy planning. The training was conducted on a single NVIDIA GeForce RTX 2080 Ti graphics card. The total training times for three different nnU-Net models were as follows: 52 h and 30 min for the 2D model, 29 h and 10 min for the 3D full-resolution model, and 50 h and 15 min for the 3D cascade model.

##### Model Selection

The final model selection was based on the highest average DSC score achieved during validation. Additionally, ensembles of the three different nnU-Net models (2D, 3D-fullres, 3D cascade) were evaluated to explore whether combining multiple models could improve performance. The 3D full-resolution configuration was selected because it outperformed the 2D and 3D cascade configuration in terms of DSC score during the validation. Henceforth, when referring to the nnU-Net model, we refer to the trained 3D full-resolution model. The median inference time for the 3D full-resolution model was 146 s per patient.

#### 2.3.2. SynthSeg

For comparison to the nnU-Net, we employed SynthSeg, a publicly available state-of-the-art segmentation model, as an example of an off-the-shelf solution. SynthSeg is also a 3D U-Net-based DL model originally trained on a large synthetic dataset (N = 1020) [18]. The segmentation process was executed using the pretrained SynthSeg model integrated into FreeSurfer (v7.3.2). As the SynthSeg model was trained exclusively on synthetic MRI scans, its application during inference was limited to T1CE MRI data in both the internal and external test sets. SynthSeg was used for the segmentation of the left and right lateral ventricles, left and right inferior lateral ventricles, and the third and fourth ventricles. These segmentations were combined to create a single total ventricle structure for each patient. Because of the model’s capacity to generalize across heterogeneous data, no additional fine-tuning or retraining was performed. Using an Intel Core i7−9700K CPU with 32 GB of DDR4 RAM (Rocky Linux 9), the median inference time per patient was 107 s.

### 2.4. Evaluation

#### 2.4.1. Segmentation Metrics

The performance of automatic ventricle segmentation methods was evaluated by comparing them to manual segmentations using several key metrics (Figure 1) in Matlab R2023a (The MathWorks Inc., Natick, MA, USA). In addition, the GT and automated segmentation volume was evaluated and compared to check for systematic differences between the segmentations in the test sets. Two widely recognized metrics, the Dice Similarity Coefficient (DSC) and 95 percentile Hausdorff distance (HD95), were used.

The DSC is a measure of volumetric overlap and is calculated as twice the overlap of the two segments divided by the total size of both volumes combined. The Hausdorff distance is defined as the largest distance between the boundary points of one contour and the nearest point in the other. The HD is computed bidirectionally, from the automated segmentation to the GT, and vice versa, and the maximum of those two directions is taken. The 95th percentile HD (HD95) is taken, as it is more robust to outliers.

However, both DSC and HD95 have been shown to have little correlation to the time needed to correct the contours and, thus, clinical contour quality [19]. Therefore, the surface DSC and added path length (APL) metrics were added, as they have been shown to correlate strongly with the number of manual corrections required of the automated contours and thus to time-saving and clinical contour quality [20].

The surface DSC was first proposed by Nikolov et al. [21] and reports the accepted surface parts (the surface parts within a tolerance of the true boundary) compared to the total surface (sum of automatic contour surface area and GT surface area). The tolerance parameter of the surface DSC represents interobserver variations in segmentations and was set to 1 mm, as brain OAR delineation interobserver variability is often around this value [22].

The APL calculates the total length of the boundary, which requires manual adjustment to meet institutional contouring guidelines. Again, a tolerance parameter of 1 mm was used to define acceptable deviation from the GT. Together, these four metrics comprehensively analyze both spatial overlap and boundary precision.

While the DSC, HD95, and surface DSC were used to evaluate the contour quality of both ventricle and PVS segmentations, the APL metric was applied exclusively to ventricle evaluation. The PVS is not a manually delineated OAR but rather an expansion of the ventricle segmentation, meaning the APL does not provide correct information for this structure.

**Figure 1 cancers-17-01598-f001:**
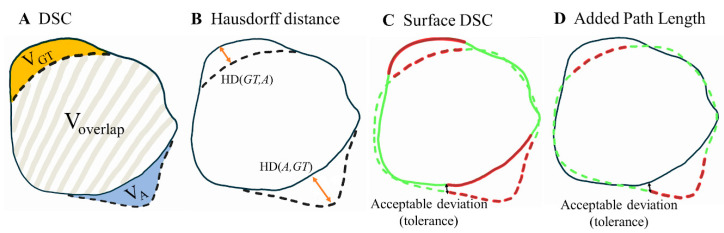
Illustration of the metrics used to evaluate the segmentations in this study, adapted from [20,21]. The solid line represents the manual ground truth segmentations (GT), while the dashed line depicts the automated segmentations (**A**). (**A**) shows the volumetric DSC, calculated as twice the overlap between two volumes (2 × V_overlap_) divided by the sum of their volumes (V_GT_ + V_A_). (**B**) illustrates the Hausdorff distance, defined as the maximum bidirectional nearest neighbor distance (orangearrows) between points on the two boundaries (max of HD(GT,A) and HD(A,GT)). (**C**) presents the surface DSC. Black arrow: the maximum deviation tolerated without penalty, set at 1 mm. Green: accepted surface parts (distance between surfaces < 1 mm). Red: unacceptable surface parts (distance between surfaces > 1 mm). The surface DSC reports the accepted surface parts (the surface parts within a tolerance of the true boundary) compared to the total surface (sum of automatic contour surface area and GT surface area). (**D**) demonstrates the added path length (APL) Green dashed line: accepted segmentation (distance between segmentations < 1 mm). Red dashed line: added path length which represents the length of the boundary in millimeters that required manual adjustment.

#### 2.4.2. Local Evaluation of Ventricle Segmentation

This study aimed to assess the variation in anatomical differences, or segmentation errors, between the automatic segmentations and the manually delineated GT. To evaluate the variation in segmentation errors across this population, it is necessary to align all ventricle contours to a common reference orientation. A standardized ventricle OAR reference shape was used following the method proposed by Brouwer et al. [23]. For this purpose, a ventricle delineation with a volume close to the median volume over all patients was visually inspected on anatomical accuracy and consequently selected as the reference shape.

The DL-based segmentations from both the nnU-Net and the SynthSeg models, as well as the manual GT contours, were converted into 3D discrete surface mesh models for each patient using the software AIQUALIS (Inpictura Ltd., Abingdon, England). These surface meshes were then registered to the reference ventricle surface mesh using a deformable shape-to-shape registration using an iterative closest point (ICP) algorithm [24]. First, a rigid ICP was used to achieve a rough initial alignment to the reference shape, followed by a non-rigid registration to fine-tune the alignment and account for the anatomical differences across the population.

The median and range (10th–90th percentiles) of the segmentation error of each vertex point of the 3D surface meshes were calculated over all patients within the test sets. Percentiles were used to be more robust to outliers within the distribution of segmentation errors for each vertex points. The conversion to 3D surface meshes, registration, and visualization of errors were all done using the software AIQUALIS (Inpictura Ltd., Abingdon, England). A low median segmentation error and low range suggest a general acceptance of the DL contours. A high median error and low range could be interpreted as a systematic error in the DL contour. A low median error and high range suggest that at that location, the segmentation model performed well in general but that there are cases in which the segmentation failed or that it is a region with high interobserver variability in the GT delineations.

#### 2.4.3. Clinical Evaluation Ventricle Segmentation

Within this study, clinically relevant contour quality is the main endpoint for which physician evaluation remains the golden standard [25]. Therefore, all three segmentations (GT, nnU-Net, SynthSeg) of all 18 patients within the two test sets were independently scored by two radiotherapy technicians. All the identifying features of the binary label map segmentation files were removed, and the files were randomized in a dataset of 108 segmentations. The two radiotherapy technicians were blinded to the segmentation origin and independently scored each of the 108 segmentations on a 4-point Likert scale: (1) Not usable; (2) Major adjustments needed; (3) Acceptable/minor adjustments; (4) Good.

### 2.5. Experiments and Statistical Analysis

The trained nnU-Net model and the SynthSeg model were applied to both the internal and external test sets to provide two automated ventricle segmentations in addition to the GT manual delineations per patient. Segmentation volumes were extracted and compared between the test sets. In addition, the corresponding PVS segmentation was extracted for both the nnU-Net and SynthSeg segmentations by taking a 4 mm margin surrounding the ventricle segmentation. The four segmentation metrics, DSC, HD95, surface DSC, and APL, were extracted to compare the nnU-Net and SynthSeg ventricle and PVS segmentations on both the internal and external test sets. Given the small sample sizes (18 results per segmentation metric), normality was assessed using the Shapiro–Wilk test, a widely accepted method for evaluating whether data follows a normal distribution, especially in small datasets. A threshold of 0.05 was used for significance to test normality. If the data were normally distributed, a t-test was used to compare the segmentation results. For non-normally distributed data, the non-parametric Mann–Whitney U test was employed to assess differences without assuming normality. All statistical tests were implemented in Matlab R2023a.

The results of the models were compared against each other on the same test sets, as well as the same model on both test sets. This resulted in 24 statistical tests per structure. To account for the multiple comparisons made, we applied a Bonferroni correction to adjust the significance level which minimizes the probability of false positives when testing multiple hypotheses. The Bonferroni adjusted significance level was therefore calculated to be $\alpha_{adjusted}=\frac{0.05}{48}\approx 0.001$ for both the t-tests and the Mann–Whitney U tests. The segmentation metric results were reported as median [range]. Boxplots were used to visualize the median scores and interquartile ranges of the DSC, HD95, surface DSC, and APL metrics.

## 3. Results

### 3.1. Ventricle Segmentation Results

Figure 2 and Table 2 display the performance of the nnU-Net and SynthSeg models on the internal and external test sets in terms of the DSC, HD95, surface DSC, and APL. The nnU-Net demonstrated significantly better performance on the internal test set compared to the external test set on all four metrics reported as median [range] (DSC = 0.93 [0.86–0.95] vs. 0.84 [0.69–0.89], *p* < 0.001; HD95 = 0.9 [0.7–2.5] vs. 2.1 [1.6–5.8] mm *p* < 0.001; Surface DSC= 0.97 [0.90–0.98] vs. 0.75 [0.63–0.84], *p* < 0.001; APL = 876 [407–1298] mm vs. 3653 [2612–5667] mm, *p* < 0.001).

Similarly, the SynthSeg model demonstrated significantly better performance on the internal test set compared to the external test set in terms of the surface DSC and APL (surface DSC = 0.84 [0.70–0.89] vs. 0.73 [0.55–0.78], *p* < 0.001); APL = 2809 [2311–3622] mm vs. 4562 [3373–5280] mm, *p* < 0.001).

Within the internal dataset, the nnU–Net outperformed the SynthSeg model significantly on all four metrics (DSC = 0.93 [0.86–0.95] vs. 0.85 [0.67–0.91], *p* < 0.001; HD95 = 0.9 [0.7–2.5] mm vs. 2.2 [1.7–4.8] mm, *p* < 0.001; Surface DSC = 0.97 [0.90–0.98] vs. 0.84 [0.70–0.89], *p* < 0.001; APL = 876 [407–1298] mm vs. 2809 [2311–3622] mm, *p* < 0.001). In contrast, no significant differences between metrics were observed when comparing the models on the external test set.

### 3.2. Local Evaluation of Ventricle Segmentations

Figure 3A demonstrates that the nnU-Net model generally showed a low median difference and a low range in the differences between the GT and across most of the ventricle structure, indicating overall acceptance of the DL-generated segmentations. However, in the left and right temporal horns of the lateral ventricles, a low median difference paired with a high range suggests substantial interobserver variability in the differences between GT and nnU-Net segmentation.

Figure 3B displays that for the SynthSeg model on the internal test set, a low median difference with a low range was observed, further indicating general acceptance of the automated segmentations. However, a region of low median difference with a high range is apparent in the middle of the left and right temporal horns of the lateral ventricles, as well as in the occipital horn of the lateral ventricles, which again points to relatively high interobserver variability in the differences between the GT and the SynthSeg contours. An example of an internal test set patient where both the nnU-Net and the SynthSeg model failed to segment the central region of the left and right temporal horn of the lateral ventricles is displayed in Figure 4A.

Figure 3C displays that the nnU-Net model showed a high median difference and a wide range in the central region of the left and right temporal horns of the lateral ventricles, suggesting that the model frequently failed to accurately segment this region on the external test set. Additionally, the occipital horns of the lateral ventricle and the inferior region of the third ventricle exhibited a low median difference with a high range, suggesting substantial interobserver variability in the differences between GT and nnU-Net segmentation for this anatomical region.

Figure 3D displays that on the external test set, where the SynthSeg model has a low median and high range in the central region of the left and right temporal horns of the lateral ventricles, as well as in the caudal part of the third ventricle and the cerebral aqueduct. The low median and high range suggest substantial interobserver variability in the differences between GT and SynthSeg segmentations for these anatomical regions. In addition, Figure 3D displays a high median difference and a wide range in the occipital horn of the lateral ventricles, suggesting that the SynthSeg model often failed in segmenting this region accurately.

Across all models and datasets, we consistently observed that the lateral apertures of the fourth ventricle had high ranges and often high median differences between the GT and the automatic segmentations. This suggests that the nnU-Net and SynthSeg models failed to accurately segment this structure in both test sets.

### 3.3. Clinical Evaluation Ventricle Segmentation Results

Table 3 displays the results of the clinical rating of the ventricle segmentation. The technicians individually scored the segmentations with similar results. No significant differences were found between their ratings. However, the rating of the GT delineations differed significantly between the internal and external test sets with a score of 3.5 [2.7–4.0] vs. 2.9 [2.6–3.2], *p* < 0.001, respectively.

On the internal test set, the nnU-Net achieved significantly higher ratings than the SynthSeg model (3.8 [3.3–4.0] vs. 2.6 [2.0–3.2], *p* < 0.001). Additionally, the GT segmentations also received a significantly higher score than the SynthSeg model on the internal dataset (3.5 [2.7–4.0] vs. 2.6 [2.0–3.2], *p* < 0.001). Lastly, the nnU-Net received a higher clinical rating compared to the GT on the internal test set, although this difference was not significant.

In the external dataset, the nnU-Net also received significantly higher scores than the SynthSeg model (3.6 [3.0–4.0] vs. 2.6 [2.0–3.1], *p* < 0.001). In addition, the nnU-Net received a significantly higher score compared to the GT in the external dataset (3.6 [3.0–4.0] vs. 2.9 [2.6–3.2], *p* < 0.001).

### 3.4. PVS Segmentation Results

Table 4 and Figure 5 display the PVS segmentation performance of the nnU-Net and SynthSeg models on the internal and external test sets in terms of DSC, HD95, and surface DSC. In terms of the differences between the PVS segmentations resulting from the automated ventricle segmentations, the nnU-Net had a significantly higher DSC and surface DSC in the internal test set compared to the external set (DSC = 0.87 [0.82–0.89] vs. 0.76 [0.69–0.79], *p* < 0.001; Surface DSC = 0.85 [0.76–0.91] vs. 0.80 [0.69–0.83], *p* < 0.001). In addition, the volume of the segmentation was significantly higher in the internal dataset (86.8 cm^3^ [63.6–110.0] vs. 59.7 cm^3^ [52.3–75.1], *p* < 0.001).

The SynthSeg model had a significantly higher DSC in the internal test set compared to the external test (DSC = (0.80 [0.72–0.83] vs. 0.74 [0.64–0.78], *p* < 0.001). In addition, the segmented volume was significantly higher in the internal test set compared to the external set (85.5 cm^3^ [60.1 –106.7] vs. 59.1 [46.1–75.4], *p* < 0.001)

When comparing the nnU-Net to the SynthSeg model on the internal test set, the nnU-Net showed a significantly higher DSC and surface DSC (DSC = 0.87 [0.82–0.89] vs. 0.80 [0.72–0.83], *p* < 0.001; surface DSC = 0.85 [0.76–0.91] vs. 0.73 [0.29–0.77], *p* < 0.001). In contrast, no statistically significant differences between metrics were found between the nnU-Net and the SynthSeg model on the external test set.

## 4. Discussion

This study developed a DL model to segment the ventricles and PVS automatically, following EPTN guidelines on CT and T1CE MRI in a neuro-oncological patient population. The model’s performance was then assessed against an off-the-shelf segmentation model.

We observe that in terms of segmentation metrics, both models seemed to perform better on the internal test set compared to the external test set. This may be expected from the nnU-Net, as both the training and internal test set were sourced from the same patient population and delineated by the same two individuals. For the SynthSeg model, it was surprising that it performed significantly better in terms of the surface DSC and APL on the internal test set compared to the external test set, as it was originally trained on a different dataset, and no preference was expected.

The clinical rating shows no significant differences in segmentation quality between the nnU-Net and SynthSeg models across internal and external test sets. However, discrepancies between the clinical ratings and segmentation metrics are well documented [26,27,28]. Kofler et al. [29] found that the DSC, HD, and surface DSC had only moderate correlation with clinical ratings, with HD showing particularly weak, and sometimes even negative, correlation. These findings highlight the need for new segmentation metrics with better correlation to clinical ratings.

Despite following the same EPTN guidelines, the clinical rating revealed significant differences in the GT delineations between the two test sets. While these guidelines help standardize OAR delineation and reduce interobserver variability, studies show they do not eliminate it entirely [4].

Within the internal test set, the nnU-Net outperformed the SynthSeg model across all four segmentation metrics. Notably, the nnU-Net’s significantly better surface DSC and APL scores on the internal test set emphasize its advantage over SynthSeg, as they are linked to clinical time savings in delineation [20]. Additionally, clinician ratings significantly favored the nnU-Net segmentations over the SynthSeg segmentations on the internal test set, suggesting greater clinical acceptability.

In contrast, no significant differences in segmentation metrics were observed between the models on the external test set, though clinician ratings again favored the nnU-Net segmentation, significantly surpassing both the SynthSeg and even GT segmentations. Similarly, previous studies [28,29] found that clinical ratings of automated CNN segmentations were consistently higher than those of their expert-created reference labels. These findings highlight that it is important to reflect upon whether the human reference labels can qualify as a GT while also highlighting the need to develop better annotation procedures. In addition, it suggests that autosegmentations have the potential to reduce interobserver variability.

The clinical ratings suggest that nnU-Net segmentations may hold higher clinical value in both an internal and external setting while also pointing to possible systematic differences in GT delineation between internal and external test sets. The lack of significant segmentation metric differences between both models in the external set may also be due to the relatively modest training set size for nnU-Net (n = 78), which may limit its ability to generalize across institutions. Additionally, both the internal (n = 18) and external (n = 18) test sets are relatively small, which may constrain the statistical power and generalizability of the findings. While the SynthSeg model was trained on a considerably larger, but synthetic, dataset (N = 1020), expanding the nnU-Net training data with additional multicenter samples could enhance its robustness and external validity. However, creating multicenter datasets for this purpose presents significant logistical challenge. Similarly, incorporating additional imaging modalities might benefit certain applications, but it falls outside the scope of this study, which already leverages the most clinically relevant and available imaging modalities for radiotherapy patients. Despite these limitations, the significantly higher clinician ratings in the external test set compared to the SynthSeg model suggest that even in its current form, nnU-Net may offer superior clinical acceptability for automated ventricle segmentation.

It is important to note that the nnU-Net was trained on both CT and T1CE MRI data, and both modalities were used as input during inference. In contrast, the SynthSeg model was trained exclusively on synthetic T1-weighted MRI scans and applied only to the T1CE MRI data in this study. While this difference reflects the practical limitations of available models, our primary aim was to compare the performance of these models as they would be used in a clinical setting. Training the nnU-Net with CT and MRI mirrors the real-world scenario where both modalities are accessible for treatment planning.

The analysis of the local segmentation error demonstrated an overall acceptance of the nnU-Net and SynthSeg segmentations across the ventricle structure. However, in the central region of the left and right temporal and occipital horns of the lateral ventricles, both models showed a low median paired with a high range difference between the automated segmentations and the GT. This pairing of low median difference and high range suggests substantial variability in the segmentation errors of both the nnU-Net and SynthSeg models within the population.

The temporal and occipital horns of the lateral ventricles, in particular, presented challenges due to their narrow size which can come close to the voxel resolution, making them difficult to detect on T1CE MRI images. Because of the sub-voxel dimensions of these structures, voxel-based segmentation methods such as those employed by nnU-Net and SynthSeg may struggle to achieve optimal accuracy. Similar challenges with disconnected tubular structures have been observed in other cranial OARs, such as the optic chiasm and optic nerves. For instance, Mlynarski et al. [30] addressed such disconnections by applying a graph-based post-processing algorithm to enforce anatomical consistency, successfully restoring connectivity between the eye and chiasm. Moreover, Shit et al. [31] proposed the soft centerline Dice (soft-clDice) loss function, which enhances both the connectivity and DSC of tubular structures by promoting topological preservation.

Both the graph-based approach by Mlynarski et al. [30] and the soft-clDice loss function by Shit et al. [31] could potentially improve segmentation accuracy and connectivity in the temporal and occipital horn regions of the lateral ventricles. However, if these connections are barely visible or entirely absent on T1CE MRIs, there may be limited clinical value in enforcing their continuity.

To contextualize the performance of the nnU-Net and SynthSeg models in our study, we compared our segmentation results to prior DL studies focused on ventricular segmentation. The reported Dice Similarity Coefficients (DSCs) in these studies range from 0.89 to 0.97. However, direct comparisons are complicated due to differences in patient cohorts, imaging modalities, and segmentation goals. For example, Shao et al. [9,10] focused on normal pressure hydrocephalus (NPH) patients, whose enlarged ventricles yielded higher contrast and simplified delineation. Their DL models trained on mixed datasets of healthy controls and NPH patients achieved DSCs of 0.97 on internal test sets, with a lower performance of 0.90 reported in a healthy cohort. However, because both studies included a substantial number of NPH patients in their test sets whose enlarged ventricles are easier to segment and presented a much larger target volume, the DSC scores were automatically skewed toward higher values. Similarly, Atlason et al. [11] developed a DL model for ventricle parcellation across multimodal MR sequences (T1, FLAIR, and T2), reporting a mean DSC of 0.91 in a mixed pathology dataset with a large range of ventricle sizes. In contrast, Ntiri et al. [12] achieved a DSC of 0.96 using their DL network on 501 T1-weighted MRIs from older adults with various cerebrovascular lesions. However, they explicitly acknowledged the lack of brain tumor patients in their training set, limiting its relevance to oncologic populations. In our study, the nnU-Net achieved a DSC of 0.93 on the internal dataset, which is comparable to these benchmarks despite the different clinical context. Both the nnU-Net and SynthSeg models achieved lower DSCs (0.84) on an external dataset, highlighting a performance drop that may be expected when transitioning from internal to unseen multi-institutional data. Taken together, while our DSC values are slightly lower than some published scores, this may be attributable to a more clinically complex patient population (brain tumor patients with potentially distorted ventricle anatomy) and the use of external validation.

A study by Lorenzen et al. [22] looked at the interobserver variability in the delineation of OARs in the brain. They showed that the standard deviations between delineations were below 1 mm in most OARs. In addition, the interobserver DSC scores for organs of comparable size to the ventricles, like the brainstem, were around 0.89. Moreover, a study by Nielsen et al. [32] showed similar standard deviations and DSC scores for several OARs in head and neck cancer patients. Within the internal test set, the high surface DSCs of the nnU-Net and the SynthSeg model, together with the median and 10–90 percentile range images, show that the models have comparable results to interobserver variability for brain OARs delineations. However, interobserver variability for the EPTN 2021 structures, which includes the ventricles, is not known. Investigating this variability could be a valuable direction for future research, particularly given the differences observed between the internal and external datasets in this study.

In the PVS segmentation task, our results highlight the impact that errors in the ventricle segmentation can have on the accuracy of the resulting PVS segmentations. We also observed a general trend of lower DSC and surface DSC and higher HD95 in the PVS segmentation in comparison to the underlying ventricle segmentations. Since the PVS is defined by the region surrounding the segmented ventricles, inaccuracies in the ventricle boundaries propagate into the PVS, affecting both the DSC and surface DSC metrics. This cascade underscores the need for accurate ventricle segmentations to ensure precise PVS delineations, especially in radiotherapy where the dose constraints apply to the PVS rather than the ventricles.

Nevertheless, previous studies suggest that contouring variability only affects dosimetric outcomes when the OAR edge lies near a high dose gradient. In most cases, contouring variability does not significantly influence dosimetric outcomes [33,34]. These finding suggest that despite differences in PVS segmentation accuracy, the performance of the nnU-Net and SynthSeg models may suffice for radiotherapy planning applications. When the OAR edge is not near a high-dose gradient, additional contour precision is unlikely to produce meaningful dosimetric benefits

Furthermore, emerging studies suggest that the PVS may play a significant role in the development of radiation-induced contrast enhancement (RICE). For instance, Bahn et al. [2] demonstrated that the PVS is a predisposition site for RICE occurrence and utilized PVS segmentation as an input for their RICE prediction model. While direct evidence on the clinical benefits of PVS sparing is currently limited, Lütgendorf-Caucig et al. [35] reported that RICE near the PVS was associated with transient declines in cognitive function and health-related quality of life at 12 months post-treatment, although these declines diminished in the long term. These observations highlight that minimizing unnecessary radiation exposure to the PVS may offer short-term cognitive benefits for patients undergoing radiotherapy. Moreover, the ability to reliably segment the PVS through robust ventricle segmentation models holds significant value for research, particularly in advancing predictive models of treatment-related side effects.

While this study primarily focused on radiotherapy planning, the developed model could also assist in monitoring ventricular size, potentially aiding hydrocephalus management in a similar manner to the work by [9,10]. Although we did not explicitly validate volume tracking, the model’s accuracy in segmenting these structures suggests it could be useful for longitudinal tracking. Future work could further explore this potential, especially with longitudinal datasets, to assess its efficacy in volume monitoring and for hydrocephalus.

The nnU-Net model has demonstrated strong performance in this study, with accuracy and consistency in both research and clinical applications. However, successful integration into clinical workflows requires addressing several challenges. The process typically involves a two-step framework: commissioning, which includes training, validation, and testing (already completed in this study), and implementation and quality assurance, which focuses on embedding the model into clinical workflows [36]. The model’s fast inference time (under 3 min) shows promise in improving workflow efficiency. Crucially, the superior segmentation accuracy and clinical acceptability of nnU-Net can translate directly into time savings during treatment planning by reducing the need for manual corrections. This is particularly valuable in the case of complex structures like the PVS, where manual delineation is time-consuming.

To ensure reliable clinical use, the model’s outputs must meet clinical standards, and outliers should be addressed through continuous quality assurance. Additionally, training clinical users on the model’s capabilities and limitations is essential. With these considerations in place, the insights gained from this study can be applied to integrate nnU-Net into clinical practice, offering significant benefits in terms of segmentation accuracy, consistency, and overall workflow efficiency in clinical radiotherapy.

## 5. Conclusions

A deep learning-based (nnU-Net) model for the automatic segmentation of the ventricles and PVS according to the EPTN 2021 guidelines was developed and validated. The model showed good performance across both internal and external test sets, with segmentation results falling within the interobserver delineation variability. On the internal dataset, the nnU-Net model surpassed an openly available off-the-shelf model (SynthSeg) in terms of both the segmentation metrics and clinical acceptability scores. In the external test set, while no significant differences in the segmentation metrics were observed between the two models, the nnU-Net model received significantly higher clinician ratings, indicating a potential advantage in the clinical acceptability of the ventricle segmentations. Therefore, training a model from scratch may be considered if there is a need for specialized fine-tuning to address institution-specific data characteristics. Otherwise, selecting an existing pretrained model would likely yield comparable outcomes, making it a practical and efficient alternative. The developed nnU-Net model, made publicly available on Cancerdata.org and GitLab, holds promise for improving radiotherapy planning workflows by reducing manual segmentation efforts and could also support future research in predicting treatment-related side effects.

## Figures and Tables

**Figure 2 cancers-17-01598-f002:**
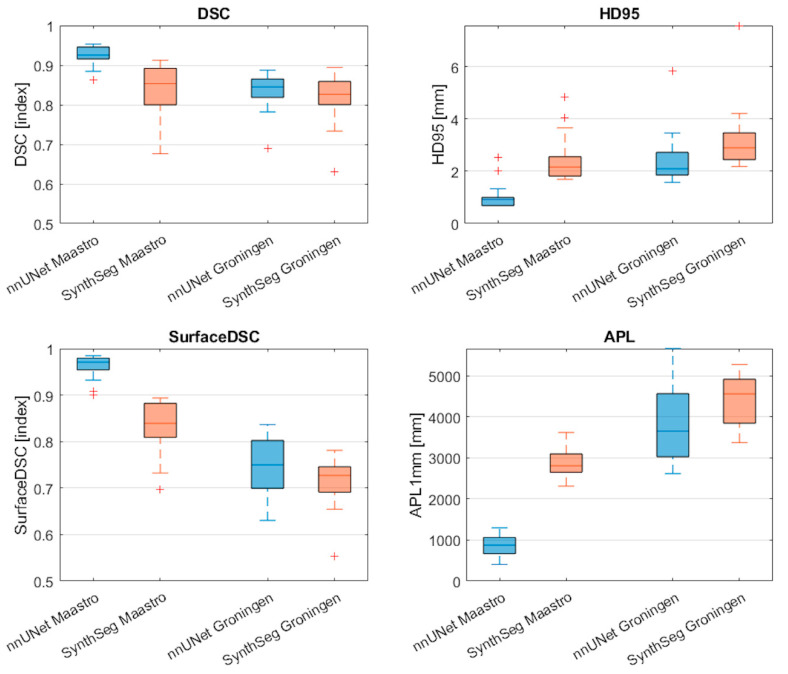
Comparison of segmentation performance of the nnU-Net and SynthSeg on the internal test set and external test set. Red cross markers represent outliers (>1.5 interquartile range). DSC: Dice Similarity Coefficient; HD95: Hausdorff distance 95th percentile; APL: added path length.

**Figure 3 cancers-17-01598-f003:**
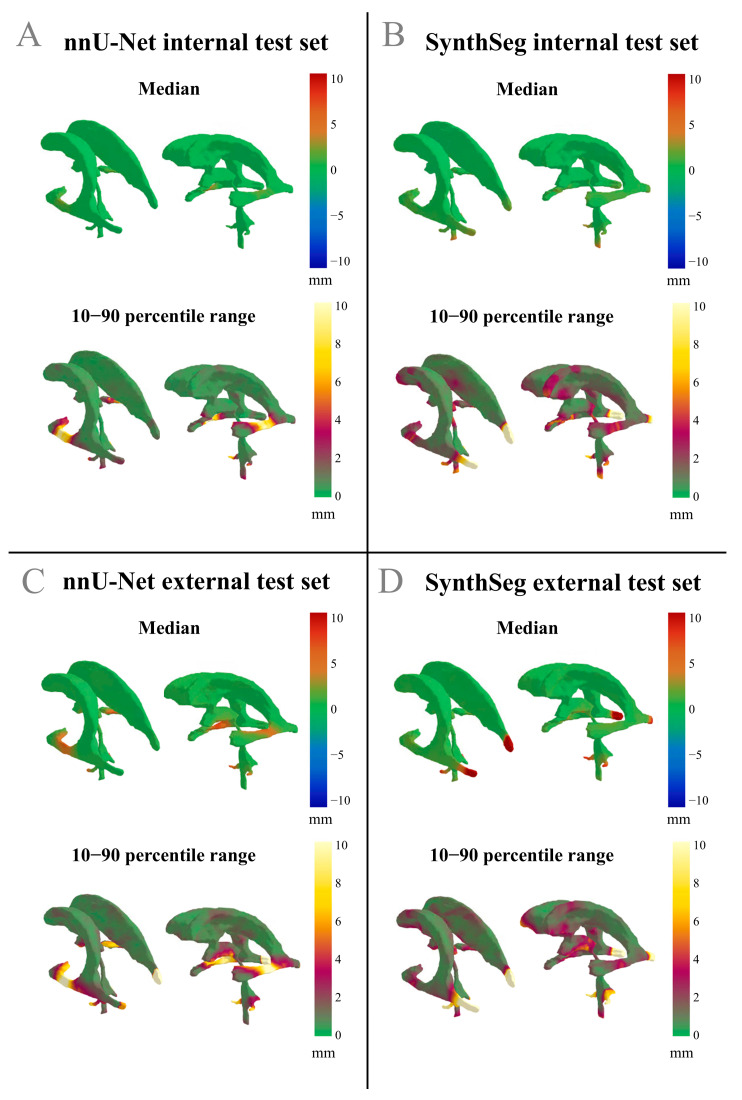
Differences in median values and the 10–90 percentile range displayed on the reference shape of the ventricles. Positive differences indicate an outward deviation of the automated segmentation compared to the GT. The 10–90 percentile range illustrates the variability of the differences between the automated segmentations and the GT across a patient population. (**A**) nnU-Net results from the internal test set; (**B**) SynthSeg results from the internal test set; (**C**) nnU-Net results from the external test set; (**D**) SynthSeg results from the external test set.

**Figure 4 cancers-17-01598-f004:**
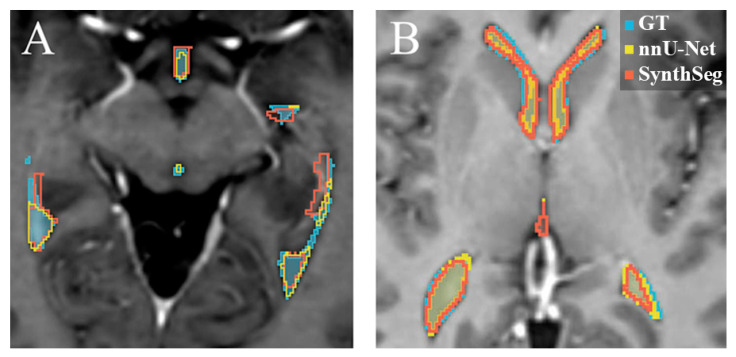
Examples of the automatic segmentations overlayed on transversal T1-weighted post-contrast MRI slices. Blue: GT manual delineation. Yellow: the nnU-Net segmentation. Red: the SynthSeg segmentation. (**A**) is an example where both the nnU-Net and the SynthSeg model failed to segment the lateral arm of the ventricle correctly on the internal test set. (**B**) displays an example of the segmentation on a patient in the external test set, where the nnU-Net and the SynthSeg model yielded similar results.

**Figure 5 cancers-17-01598-f005:**
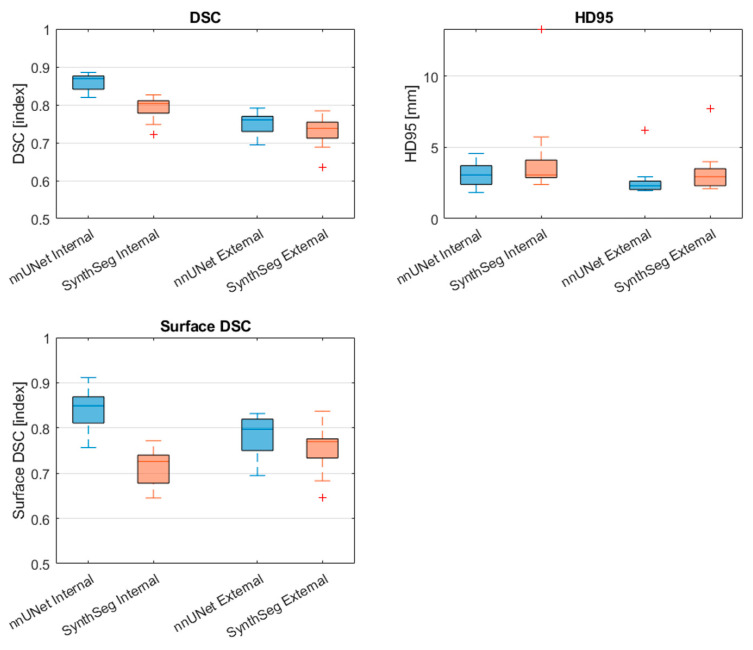
Comparison of PVS segmentation performance of the nnU-Net and SynthSeg models on the internal test set and external test set. Red cross markers represent outliers (>1.5 interquartile range). DSC: Dice Similarity Coefficient, HD95: Hausdorff distance 95th percentile, Surface DSC: surface Dice Similarity Coefficient.

**Table 1 cancers-17-01598-t001:** Patient characteristics internal and external datasets.

	Internal Training Set (n = 78)	Internal Test Set (n = 18)	External Test Set (n = 18)
Sex			
- Male	33 (42%)	11 (61.1%)	14 (77.8%)
- Female	45 (58%)	7 (38.9%)	4 (22.2%)
Median age (IQR); range	54.5 (40.8–68); 23–83	57.0 (41.5–63.3) 25–74	43.1 (33.0–51.1); 29–59
- Diagnosis count (%)			
- Meningioma G1	15 (19%)	2 (11.1%)	0 (0.0%)
- Meningioma G2	9 (11%)	1 (5.6%)	0 (0.0%)
- Oligodendroglioma G2	11 (14%)	1 (5.6%)	3 (16.7%)
- Oligodendroglioma G3	8 (10%)	2 (11.1%)	4 (22.2%)
- Astrocytoma G2	18 (23%)	6 (33.3%)	6 (33.3%)
- Astrocytoma G3	2 (3%)	0 (0.0%)	3 (16.7%)
- Other	15 (19%)	6 (33.3%)	2 (11.1%)
Resection	62 (79%)	15 (83.3%)	15 (83.3%)
Tumor location			
- Frontal	34 (43%)	4 (22.2%)	12 (66.6%)
- Occipital	6 (8%)	0 (0.0%)	1 (5.6%)
- Temporal	15 (19%)	4 (22.2%)	2 (11.1%)
- Parietal	12 (15%)	3 (16.7%)	2 (11.1%)
- Other	11 (14%)	7 (38.9%)	1 (5.6%)

**Table 2 cancers-17-01598-t002:** Segmentation results from the nnU-Net and SynthSeg models on the internal and external test set reported as median (range).

Internal test set	MODEL	DSC [index]	HD95 [mm]	Surface DSC [index]	APL [mm]	Volume GT [cm^3^]	Volume Segmentation [cm^3^]
	nnU-Net	0.93 (0.86–0.95)	0.9 (0.7–2.5)	0.97 (0.90–0.98)	876 (407–1298)	26.0 (9.1–52.8)	25.0 (92.7–56.9)
	SynthSeg	0.85 (0.67–0.91)	2.2 (1.7–4.8)	0.84 (0.70–0.89)	2809 (2311–3622)	26.0 (9.1–52.8)	24.1 (8.8 –60.3)
Internal test set	nnU-Net	0.84 (0.69–0.89)	2.1 (1.6–5.8)	0.75 (0.63–0.84)	3653 (2612–5667)	29.7 (15.3–56.9)	22.9 (10.0–48.5)
	SynthSeg	0.83 (0.63–0.89)	2.9 (2.2–7.6)	0.73 (0.55–0.78)	4562 (3373–5280)	29.7 (15.3–56.9)	25.0 (9.8–56.0)

**Table 3 cancers-17-01598-t003:** Results of the clinical rating of the ventricle segmentations. Segmentations were scored on a 4-point scale: (1) Not usable; (2) Major adjustments needed; (3) Acceptable/minor adjustments; (4) Good.

Segmentations	Clinical Rating, Mean ± SD
Internal dataset	
- GT	3.5 ± 0.8
- nnU-Net	3.8 ± 0.5
- SynthSeg	2.6 ± 0.5
External dataset	
- GT	2.9 ± 0.3
- nnU-Net	3.6 ± 0.6
- SynthSeg	2.6 ± 0.6

**Table 4 cancers-17-01598-t004:** PVS segmentation results of the nnU-Net and SynthSeg models on the internal and external test set reported as median (range).

Internal test set	MODEL	DSC [-]	HD95 [mm]	Surface DSC [-]	Volume GT [cm^3^]	Volume Segmentation [cm^3^]
	nnU-Net	0.87 (0.82–0.89)	3.1 (1.8–4.6)	0.85 (0.76–0.91)	77.2 (57.2–96.7)	86.8 (63.6–110.0)
	SynthSeg	0.80 (0.72–0.83)	3.1 (2.4–13.3)	0.73 (0.29–0.77)	77.2 (57.2–96.7)	85.5 (60.1 –106.7)
External test set	nnU-Net	0.76 (0.69–0.79)	2.3 (2.0–6.2)	0.80 (0.69–0.83)	78.3 (68.6–92.1)	59.7 (52.3–75.1)
	SynthSeg	0.74 (0.64–0.78)	2.9 (2.1–7.7)	0.77 (0.65–0.84)	78.3 (68.6–92.1)	59.1 (46.1–75.4)

## Data Availability

Original data from this manuscript will be made available upon reasonable request. The developed DL model will be made available on CancerData.org and can already be accessed via GitLab at [https://gitlab.com/ventricle_segmentation/nnunet-ventricle-segmentation] (accessed on 4 May 2025).

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
