# Peer review of "Deep Learning for Automated Ventricle and Periventricular Space Segmentation on CT and T1CE MRI in Neuro-Oncology Patients"

_cancers, 2025, doi:10.3390/cancers17101598_

Round 1

Reviewer 1 Report (New Reviewer)

Comments and Suggestions for Authors

The manuscript presents a nnU-Net deep learning approach for automated segmentation of ventricles and periventricular space on CT and MRI. The aim of study is the improvement of segmentation accuracy for neuro oncology patients undergoing radiotherapy by comparing nnU-Net's performance with an off-the-shelf model, SnythSeg.

Please:

  1. Expand briefly on the clinical implications of PVS sparing in radiotherapy, such as impact on cognitive outcomes.
  2. The study includes multi-center datasets , internal: 78 ps, external: 18 ps, which is relatively small. If possible enrich the dataset, if not, please mention this small sample size as a limitation of the study for generalizability of the study in discussion section.
  3. Provide a brief rationale for why the 3D full resolution nnU-Net was chosen over other configurations.
  4. Adjust figure 4 labeling to improve, clarity, particularly in distinguishing nnU-Net from SynthSeg.
  5. Highlight how nnU-Net's superior segmentation could lead to improved radiation planning efficiency in clinical workflows.
Comments on the Quality of English Language
  1. Please correct the misspelling for "Neura Network" to "Neural Network."

Author Response

Thank you for the insightful and extensive feedback. Please see the attachment.

Reviewer 2 Report (New Reviewer)

Comments and Suggestions for Authors

Dear Authors,

Thank you for the opportunity to review your manuscript, which presents a deep learning-based approach for ventricle and periventricular segmentation in neuro-oncology patients using CT and T1CE MRI. The study addresses a clinically meaningful task with potential applications in radiotherapy planning and quantitative neuroimaging.

The manuscript is generally well-structured and well-written. The methodology is clear, and the results are promising. However, I believe the manuscript would benefit from further clarification in several areas, as outlined below.

Abstract

The abstract presents the topic clearly and summarizes the study's aims, methods, and results in a structured manner. However, a few points can be improved:

  • The Dice scores are appropriately reported. It would be helpful to specify whether these are mean values across test subjects and if the standard deviations or ranges are available; including them would better reflect performance variability.
  • The last sentence about enabling quantitative analysis is good but could be strengthened. Consider specifying how this segmentation might impact patient management, such as hydrocephalus assessment, treatment planning, or outcome prediction.

Introduction

The introduction establishes the clinical context and motivation for automated segmentation of ventricles and periventricular regions in neuro-oncology patients. Overall, it is informative and well-structured. Below are suggestions for improvement:

  1. Previous works on segmentation are cited, including U-Net and its variants. While the context is generally adequate, the study's novelty should be highlighted more explicitly. How does this study differ from or improve upon previous methods (e.g., in terms of multi-modality dataset size)?
  2. The study's aim is stated, but it would benefit from being more concise and specific. It is too long. The suggested revision is:
    • This study had two main objectives. First, we aimed to develop a deep learning model for accurate automated segmentation of the ventricles and periventricular regions using CT and T1CE MRI in accordance with EPTN guidelines for radiotherapy planning. These guidelines allow the use of dose constraints on organs at risk (OARs), making precise segmentation clinically valuable. Second, we evaluated the accuracy and clinical applicability of this custom model compared to a publicly available pre-trained model, using both internal and external test datasets. This comparison highlights the potential and limitations of off-the-shelf solutions in radiotherapy planning
    • Some sentences are slightly long or passive.

Methods

The Methods section is generally clear and follows a logical structure. However, there are several areas where clarification and refinement would improve.

  1. The dataset includes CT and T1CE MRI images of neuro-oncology patients, but more detail is needed:
    • Patient demographics (e.g., age range) are not provided here.
    • Clarify whether the dataset has been collected retrospectively under IRB approval.
    • Were segmentations double-checked or consensus-reviewed?
  1. Preprocessing steps are missing:
    • Resampling or normalization (e.g., voxel size standardization, intensity scaling)?
    • How were the periventricular regions defined or delineated?
  1. The U-Net architecture is appropriate, but more specifics would improve clarity:
    • Any modifications to the architecture (e.g., number of layers, attention blocks)?
    • What input size was used?
    • Was data augmentation applied?

4 The split into training/validation/test sets is mentioned, but the method of splitting should be clarified:

    • Was it random per subject or stratified by modality?
    • Were CT and MRI models trained separately or together?

5. Please mention whether code or model weights will be publicly available; you can give the GitHub link

6. What is your view on the potential influence of “age-related anatomical variability” on segmentation performance? The internal dataset appears to have a wider age range, which may lead to “more prominent age-related ventricular enlargement.” Could this impact model performance or explain differences between internal and external datasets?

Results

The Results section presents performance metrics for the segmentation models. The results are promising, but some areas need clarification and more detailed reporting:

  1. Please specify whether these values are mean ± SD for all Dice scores across test subjects.
    • Were statistical comparisons performed between CT and MRI Dice scores? If so, report p-values
  1. The inclusion of segmentation examples is very helpful. Please add qualitative commentary in the text—for example, did the model fail in specific cases (e.g., small ventricles, which type of tumors, low contrast)?
  2. The results would benefit from a brief analysis of model limitations:
    • Were any clinical or imaging factors (e.g., edema, mass effect, shunts) associated with poor segmentation?

Discussion

The Discussion section appropriately interprets the model's performance and potential utility in clinical practice. However, it can be strengthened with more detailed comparisons to prior work, expanded clinical relevance, and a clearer acknowledgment of limitations.

  1. The authors mention potential use in quantitative analysis, but this can be made more impactful. Consider elaborating:
    • How could automated segmentation be used for hydrocephalus monitoring or treatment planning?
    • Could the model assist in longitudinal tracking of ventricular size or periventricular changes?
  1. The model's Dice scores should be compared with previous segmentation efforts, if available, especially those using ventricle segmentation or multi-modality input.
    • Was this the first study focusing on periventricular segmentation in neuro-oncology using deep learning? If yes, that should be emphasized as a novel contribution.
  1. Some limitations are briefly mentioned but deserve more depth:
  • Were imaging protocols standardized across cases, or did variability in slice thickness, orientation, or "contrast timing" impact model performance?
  • Retrospective design and potential annotation variability should also be noted.
  1. The mention of future prospective validation is welcome. Consider being more specific:
    • Is there potential for cross-modality models or domain adaptation?
    • Could adding FLAIR or T2-weighted MRI improve periventricular region delineation?

Conclusion

The conclusion restates the study's main findings but could be improved with more specificity and a stronger closing statement.

  • Consider rephrasing the final sentences to emphasize the practical implications. And include a brief statement on how the model could be translated to clinical practice or used for future research.

Language

    1. The manuscript is generally well-written and professionally presented. Minor edits can further improve readability. You can replace long passive sentences with an active voice where possible.
    2. "nnU-Net" should be spelled consistently throughout the manuscript. In some places, it appears as "nnU-net," which is incorrect.
    3. "SynthSeg" should also be used consistently—with or without "the," choose one style and maintain it throughout.
    4. There should be a space between numbers and units (e.g., write "2.1 mm" instead of "2.1mm").

References

    1. The reference list includes relevant and recent studies. Please ensure uniform formatting (e.g., DOI placement).

Author Response

We would like to thank you for the extensive and insightful feedback. Please see the attachment.

Reviewer 3 Report (New Reviewer)

Comments and Suggestions for Authors

Review for "Deep Learning for Automated Ventricle and Periventricular Space Segmentation on CT and T1CE MRI in Neuro-Oncology Patients"

Ch1.: The introduction emphasizes the importance of segmenting the ventricles and the periventricular space (PVS) in the brain. This introductory chapter lacks a broader comparison with state-of-the-art segmentation models such as nnU-Net and SynthSeg in the context of other existing ventricle or brain tissue segmentation studies. Even if PVS-specific methods are rare, comparison with general-purpose or relevant brain segmentation approaches will strengthen the positioning of this paper in the domain.

Ch2.: The method proposed is well described and comprehensible, especially the model training and evaluation metrics. However, in Section 2.4, there is a typo. The section starts with a full stop, which should be corrected. The CNN hyperparameter setup is well explained.

Ch3.: The segmentation performance is well presented across internal and external datasets. However, Table 2 lacks a comparison with performance reported in similar articles of the domain. It's difficult to evaluate these results without contextualizing the results of benchmarks from related research. Also, the comparison between the two models (nnU-Net and SynthSeg) seems too simple and should be extended with insights from other studies.

Ch4.: The discussion rightly identifies key limitations and clinical relevance but does not sufficiently compare findings to the existing ventricle or brain tissue segmentation research. The performance of nnU-Net and SynthSeg is unclear in terms of how these models compare to other recent literature, especially in the ventricle or white matter segmentation studies. A thorough study of research papers using nnU-Net or SynthSeg in the brain tissue segmentation domain would underline the relevance of the results obtained and put the findings into context.

Ch5. The conclusion summarizes the work well, emphasizing clinical relevance. The final sentence mentions the model will be published on GitHub, but no link is not present. The link should be added to permit the public access to the related source code. Besides, a Data Availability Statement should be included after the conclusion.

The are some minor typos in the article:

A comment about the authors' surnames is left at the beginning. Delete it. Section 2.4 begins with a stray full stop and should be corrected.

My overall verdict:  This is a well-written and clinically relevant article that helps in ventricle and PV segmentation. The article can be accepted after Major Review.

The manuscript should be improved significantly with the following:

-Comparative analysis with state-of-the-art segmentation papers.

- A formal data availability section.

- Fixes to the minor typographical issues mentioned above.

Author Response

We would like to thank you for the insightful and extensive feedback. Please see the attachment.

Round 2

Reviewer 3 Report (New Reviewer)

Comments and Suggestions for Authors

All the concerns formulated by my previous review have been addressed.

This manuscript is a resubmission of an earlier submission. The following is a list of the peer review reports and author responses from that submission.

Round 1

Reviewer 1 Report

Comments and Suggestions for Authors

Clarity of Purpose: The introduction effectively highlights the clinical importance of ventricle segmentation in radiotherapy. Consider elaborating more on the gap between manual delineation challenges and the current state-of-the-art automated methods to emphasize the need for this study.

Dataset Details: The dataset description is thorough, but adding a table summarizing patient demographics, imaging modalities, and key statistics might improve readability and accessibility of this information.

Statistical Analysis Explanation: While statistical methods like the Shapiro-Wilk test and Bonferroni adjustment are mentioned, including a brief rationale for their use would help readers unfamiliar with these methods understand their relevance.

Clinical Implications: The results show that nnU-Net outperforms SynthSeg in clinical ratings. It would be valuable to discuss the potential for implementing nnU-Net in clinical workflows and any barriers to adoption.

Error Analysis: The discussion of segmentation errors in specific ventricle regions (e.g., temporal and occipital horns) is insightful. Including visual examples of these errors could enhance the reader's understanding of the challenges faced by the models.

Future Directions: The conclusion briefly mentions the potential for larger datasets to improve model robustness. Expanding on this with specific suggestions, such as integrating multi-center datasets or incorporating additional imaging modalities, would provide a clearer path forward.

Comments on the Quality of English Language

The quality of English in the document is commendable, with clear and precise technical language suited for the target audience. However, some sentences are overly complex and could benefit from simplification for improved readability. Occasional inconsistencies in terminology, such as variations in referring to the "ventricular system" and "PVS," should be standardized for clarity. While abbreviations are generally well-defined, ensuring all are introduced before use will aid comprehension for a broader audience. Minor adjustments in sentence structure and the use of active voice in some sections could enhance engagement without compromising the academic tone.

Author Response

Thank you for the feedback.

Reviewer 2 Report

Comments and Suggestions for Authors

This paper proposes a method for segmenting the ventricles and periventricular space. There is sufficient significance in segmenting the ventricles using machine learning models in CT or MRI, and it seems to be an interesting research topic. Furthermore, the evaluation methods appear to be well-designed, with the adoption of APL being a noteworthy point. However, there are several significant concerns in this paper, and in its current state, it would be difficult to accept. Below are the main points of concern:

1. Lack of Novelty

The first major concern is regarding the novelty or originality of this research. The two models used for comparison in the paper are both existing methods, and there seems to be no clear innovative aspect in this study. If the main purpose of the paper is to compare existing models, it is advisable to set more rigorous comparison conditions and conduct evaluations not just on these two models, but on multiple models to provide a broader comparison.

2. Questions About Fairness in Model Comparison

The second concern, which is even more critical, is the possibility that the training datasets for the two models being compared may differ. It appears that nnU-Net might have been trained on both CT and T1CE MRI data, while the other model may have been trained only on T1CE MRI. If this is indeed the case, the comparison would not be valid in the first place. This point needs clarification. Specifically, one should clearly state the architecture of each model, how the data was fed into the model during training, and how the data was fed during inference as well.

3. Lack of Clarity Regarding Data Splitting

Lastly, the paper does not clearly explain how the internal dataset was split into training and validation sets. If randomization was performed, it is necessary to clarify which program or package was used and the procedure by which randomization was conducted. This is essential for ensuring reproducibility.

Author Response

Thank you for the feedback.
